# Amphiphilic Cationic Peptide-Coated PHA Nanosphere as an Efficient Vector for Multiple-Drug Delivery

**DOI:** 10.3390/nano12173024

**Published:** 2022-08-31

**Authors:** Fanghua Zhang, Chao Zhang, Shuangqing Fu, Huandi Liu, Mengnan Han, Xueyu Fan, Honglei Zhang, Wei Li

**Affiliations:** 1College of Chemistry and Environmental Science, Key Laboratory of Chemical Biology of Hebei Province, Laboratory of Medicinal Chemistry and Molecular Diagnosis of the Ministry of Education, Institute of Life Science and Green Development, Hebei University, Baoding 071002, China; 2Department of Life Science, Hengshui University, Hengshui 053000, China; 3Tianjin Key Laboratory of Molecular Optoelectronic Sciences, Department of Chemistry, School of Science, Tianjin University, Tianjin 300072, China

**Keywords:** amphiphilic core–shell, polyhydroxyalkanoates, 5-fluorodeoxyuridine, curcumin, synergetic therapy

## Abstract

Amphiphilic core–shell (ACS) nanoparticles are gaining increasing research interest for multi-drug delivery in cancer therapy. In this work, a new cationic peptide-coated PHA nanosphere was prepared by self-assembly of a hydrophobic core of biodegradable poly (3-hydroxybutyrate-co-3-hydroxyhexanoate) (PHBHHx) and a hydrophilic shell of fusion proteins of PHA granule-associated protein (PhaP) and cationic peptide RALA through a strong hydrophobic effect. The hydrophobic drug curcumin (Cur) was encapsulated in PHBHHx nanoparticles. The chemotherapy drug 5-fluorouracil (5-FU) was administered in the form of its metabolite oligomeric 5-fluorodeoxyuridine (FUdR). Fifteen consecutive FUdR (FUdR_15S_) were adsorbed on the surface of PHBHHx nanoparticles by electrostatic interaction with RALA to form Cur@PHBX-PR/FUdR_15S_. Such amphiphilic cationic nanospheres had 88.3% EE of Cur and the drug loading of Cur and FUdR were 7.8% and 12.1%. The dual-drug-loaded nanospheres showed a time-differential release of Cur and FUdR. In addition, Cur@PHBX-PR/FUdR_15S_ exhibited excellent anticancer activity and played a vital role in promoting the synergistic effect of FUdR and Cur in gastric cancer cells. The exploration of antitumor mechanisms demonstrated that Cur improved the activity of apoptosis-related proteins and cancer cells sensitized to FUdR. This amphiphilic core–shell system can serve as a general platform for sequential delivery of multiple drugs to treat several cancer cells.

## 1. Introduction

The combination therapy approach, through drug/drug, drug/gene, or gene/gene, is regarded as a viable alternative to traditional chemotherapies for drug-resistant tumor cells [1,2]. Extensive efforts have been made to develop multifunctional nanocarriers for delivery of different types of drugs [3]. The amphiphilic core–shell (ACS) nanoparticle is a diverse class of core–shell nanoparticles which have demonstrated excellent properties for co-delivery of multiple drugs, including chemotherapeutic drugs, nucleic acids, and proteins [4]. To date, many researchers have explored different ACS nanoparticles utilizing different polymer systems with their unique characteristic features [5,6,7,8]. Li has developed a co-delivery system for the treatment of multi-drug-resistant cancer cells, utilizing the unique structure of amphiphilic gelatin/camptothecin@calcium phosphate–doxorubicin (AG/CPT@CaP–DOX) nanoparticles to achieve the synergistic effect of a hydrophobic drug (camptothecin, CPT) and a hydrophilic drug (doxorubicin, DOX) [9]. Wang et al. designed a hyaluronic acid (HA)-decorated polyethylenimine-poly(d,l-lactide-co-glycolide) (PEI-PLGA) nanoparticle system for targeted co-delivery of TRAIL plasmid (pTRAIL) and gambogic acid (GA) in triple-negative breast cancer (TNBC) therapy [10]. Recently, ACS nanoparticles consisting of an outer PEI shell and a PMMA core were devised for effective delivery of miRNA-loaded plasmid to Kupffer cells (KCs), which had high cellular transfection efficiency [11]. In another study, Kang et al. designed and developed a liposome template PEI hydrogel nanoparticle for the efficient targeted delivery of CRISPR/Cas9 to tumors located in brain. The core of the ACS nanoparticle consisted of PEI hydrogel to encapsulate the Cas9 protein, and the shell consisted of 1,2-dioleoyl-3-trimethylammonium-propane chloride salt (DOTAP) lipids for efficient delivery of the polo-like kinase 1 (PLK1) gene [12]. These multifunctional ACS nanoparticles are proven to be an efficient and versatile platform for co-delivery of chemotherapeutic agents and therapeutic DNA in combinatorial therapy. However, the design and development of ACS nanoparticles with high biocompatibility, higher drug loading capacity, and controlled and sustainable drug releases still need to be improved [13,14].

Polyhydroxyalkanoates (PHAs) are natural aliphatic polymers with biocompatibility and biodegradability, and the degraded products of PHAs including oligomers and monomers are non-toxic to the cells and tissues [15]. So far, no research has been reported to have observed a carcinogenesis effect induced by PHAs when applied in medicine. Poly (3-hydroxybutyrate-co-3-hydroxyhexanoate) (PHBHHx), as a member of the PHAs family, shows good mechanical and processing properties [16]. PHBHHx has been employed in the controlled release of antibacterial, anticancer, anti-inflammatory, and antithrombotic drugs [17]. PHA can also combine with PHA granule-associated protein (PhaP), a low-molecular-weight amphiphilic protein, to assemble microspheres with core–shell structures through a strong hydrophobic interaction [18]. The surface of microspheres displays other functional proteins or peptides, which are achieved by the fusion expression of PhaP and those proteins [19]. By fusing human acidic glycoprotein (hAGP) and human epidermal growth factor (hEGF) with PhaP, PHA nanoparticles (NPs) modified with hAGP–PhaP and hEGF–PhaP have been constructed, realizing the targeted aggregation of hepatoma cells and macrophages [20]. In addition, the fusion expression of human immune costimulatory molecule B7-2 and PhaP is observed on the surface of PHBHHx NPs to provide functionalized NPs with biological activity of activating lymphocytes [21]. In view of this, we intend to design a fusion protein of cationic peptide and PhaP to modify the surface of the PHBHHx particle to form a cationic peptide-coated polymer nanosphere, which also has amphiphilic properties.

Many kinds of cationic peptides, such as GALA, CADY, KALA, RAWA, and RALA [22], can be used to coat polymers. RALA, as a typical cationic transmembrane peptide, consists of 30 amino acids, and includes 7 arginine and 6 leucine in the main chain. The presence of arginine makes RALA easily adsorb negatively charged nucleic acid drugs and form NPs spontaneously through electrostatic interaction, whereas the presence of leucine facilitates RALA to easily cross the cell membrane [23]. As a carrier, RALA improves the stability of nucleic acid drugs and increases the uptake of drugs by cells. At present, RALA has been successfully applied in the delivery and release of various functional nucleic acid drugs, such as small interfering RNAs [24], microRNAs [25], CRISPR gene-editing tools [26], and 5-fluorodeoxyuridine (FUdR) chains [27]. In our previous study, RALA has been designed for delivering the oligomeric FUdR (metabolite of 5-FU) strand for the therapy of HER2-overexpressing gastric cancer cells. The nanodrug shows good stability in simulated internal environments and exhibits excellent antitumor activity on N87 cells [28]. Hence, we attempt to construct the fusion protein of the cationic peptide RALA and PhaP, which can be self-assembled with PHA particles to form a RALA-coated PHA nanosphere. The shell of nanospheres shows a good ability to adsorb negatively charged nucleic acid drugs. The core hydrophobic region is capable of encapsulating hydrophobic therapeutic drugs, thus expanding the function of cationic polymer carriers for the single delivery of nucleic acid drugs and realizing the co-delivery of nucleic acid drugs and hydrophobic drugs.

In this work, we devise and prepare a cationic RALA-coated PHA nanosphere to co-load nucleic acid drugs and hydrophobic drugs (Figure 1). The FUdR strand containing 15 consecutive FUdR (FUdR_15S_) is used as a negatively charged nucleic acid drug [29]. Curcumin (Cur) is selected as the hydrophobic drug due to its non-toxicity and the ability to sensitize cancer cells to FUdR [30]. Cur is encapsulated in PHBHHx particles by emulsification/solvent evaporation to form a hydrophobic core of Cur@PHBX. The fusion protein of PhaP–RALA (PR) is purified from the constructed engineered *Escherichia coli* (*E. coli*). Subsequently, the hydrophilic fusion protein and hydrophobic core self-assemble into amphiphilic cationic nanospheres (Cur@PHBX-PR) under a strong hydrophobic effect. The FUdR_15S_ is then adsorbed on the surface of Cur@PHBX-PR via electrostatic interaction with RALA, thus forming Cur@PHBX-PR/FUdR_15S_. In addition, the morphology, size, and stability of Cur@PHBX-PR/FUdR_15S_ are characterized. The drug loading, drug release, and cytotoxicity are evaluated in vitro, and the antitumor mechanism is explored. RALA-coated PHA nanospheres may serve as a platform for the co-delivery of other nucleic acid drugs and hydrophobic drugs.

## 2. Materials and Methods

### 2.1. Materials

FUdR was purchased from Alfa Aesar (Shanghai, China). FUdR15 strands were synthesized by Sangon Biotech Co., Ltd. (Shanghai, China). PHBHHx was purchased from Bluepha (Beijing, China). Rhodamine B (Rho) and Cur were purchased from Sigma-Aldrich (Shanghai, China). DAPI and CytoPainter LysoRed Indicator Reagent (ab176828) were obtained from Abcam Co., Ltd. (Cambridge, MA, USA). Trypsin, 3-(4,5-dimethylthiazol-2-yl)-2,5-diphenyl-2H-tetrazolium bromide (MTT), antibiotic-antimycotic (100×), and fetal bovine serum (FBS) were purchased from Wisent Biotechnology Co., Ltd. (Nanjing, China). All chemicals were used without further purification. Primary antibodies to Bcl-2 (bsm-33047M), Bax (bsm-33283M), and β-actin (bsm-33036M) were purchased from Bioss Antibodies. Caspase 3, 8, and 9 activity detection kits were purchased from BestBio Biotechnology Co., Ltd. (Shanghai, China). Amicon ultracentrifuge filters were purchased from Merck Millipore Ltd. (Darmstadt, Germany).

### 2.2. Preparation of Drug-Loaded PHBX NPs

Cur-loaded PHBX NPs were prepared using the modified emulsification/solvent evaporation method [31]. Briefly, 50 mg PHBHHx and 2 mg Cur were added into 8 mL and 2 mL chloroform, respectively, and the solutions were stirred to ensure the dissolution of PHBHHx and Cur. The two organic solutions above were mixed and slowly dropped into 20 mL of 1% (*w*/*v*) PVA under sonication. Then, 0.5 mL Tween-20 was added into the mixture. Subsequently, the double emulsion was sonicated using a probe sonicator at 95% power (Sonics & Materials, Newtown, CT, USA) for 1 h, and then chloroform was removed by volatilization at 35 °C. Cur-loaded PHBX NPs were collected by centrifugation at 12,000 rpm for 20 min and washed thrice with phosphate-buffered saline (PBS) solution. PHBX NPs and Rho-loaded PHBX NPs were synthesized using the same method.

### 2.3. Synthesis, Purification, and Characterization of Protein

The sequence of the fusion protein PhaP–RALA molecule used in this study was METKPYELVDAFWKNWSQSLSLFSSAGKQLEQLTLETLKQQQDALHKLTSGVDELEKELQQLTAQFNNQYTDYVKQLTGNSLNDQINEWQDKWKELSAHMHQLTVSPTKTSLSILTQTSGQFEETTKQFIEQQQLQREEAQKQLEGFLEEFKSKQLELAKKFEENSKNLFTSIKGGGSGGSGMWEARLARALARALARALARALARALRACEA.

First, the nucleic acid sequence of PhaP–RALA was inserted into the expression vector pCold I, and the recombinant plasmid was verified by DNA sequencing. The resulting plasmid was transformed into *E. coli* BL21 (DE3). The recombinant *E. coli* BL21 (DE3)/pCold-*phap*-*rala* was cultured at 37 °C in LB containing 0.5 mM ampicillin. When the optical density of the culture at 600 nm reached 0.6, IPTG was added to a final concentration of 0.1 mM, and cultivation was continued for another 14 h at 15 °C. Cells were harvested by centrifugation (8000× *g*, 4 °C, 10 min), washed twice with 1 × PBS (pH 7.4), and lysed by sonication using an ultrasonic oscillator (Sonic Materials, Newton, CT, USA). The cell debris was removed by centrifugation (8000 rpm, 4 °C, 10 min), and the supernatant was filtered through 0.45 μm filters. Subsequently, the crude protein was applied to the HisTrap HP affinity column (GE Healthcare, Piscataway, NJ, USA). The collected enzyme solution was desalted by prepacked column HisTrapTM desalting by using 1 × PBS (pH 7.4). The purified enzyme was evaluated through the Coomassie brilliant blue staining of SDS-polyacrylamide gel electrophoresis gels.

EGFP (Accession No. AFA52653) and fusion protein PhaP–EGFP (PE) were prepared using the above method.

### 2.4. Self-Assembly of Cur@PHBX-PR/FUdR_15S_

Cur@PHBX (1 mg) was added into PhaP–RALA (1.2 mg/mL) PBS buffer solution and stirred at 4 °C and low speed overnight. After centrifugation at 12,000 rpm and 4 °C for 5 min, sediments of Cur@PHBX-PR were collected. Precooled PBS solution (1 mL) was added into the precipitate, and the precipitate was slightly resuspended. Then, 5 μL FUdR_15S_ (100 μm) was added to the suspension and stirred at 4 °C and low speed for 4 h. Finally, the mixture was centrifuged at 12,000 rpm and 4 °C for 5 min to collect Cur@PHBX-PR/FUdR_15S_.

### 2.5. Characterization of Cur@PHBX-PR/FUdR_15S_

Cur@PHBX-PR/FUdR_15S_ (300 μL) was diluted 10 times with PBS and transferred into a 5 mL colorimetric dish. The zeta potential was analyzed by dynamic light scattering (DLS). Then, 10 μL Cur@PHBX and Cur@PHBX-PR/FUdR_15S_ (N:P 10) samples were deposited onto the surface of the silicon wafer, dried overnight, and observed by scanning electron microscopy (SEM). Samples were also deposited onto the surface of 200-mesh copper mesh and dried. Negative staining was carried out with 1% negative staining solution (phosphotungstic acid, pH 7.4), washed with distilled water thrice, and dried overnight for transmission electron microscopy (TEM). The Rho@PHBX-EGFP solution (1 mL, 0.2 mg/mL) was spread on the confocal dish and dried overnight, and the fluorescence image was obtained by laser confocal scanning electron microscopy (LCSE).

### 2.6. Drug Loading (DL) of Cur@PHBX-PR/FUdR_15S_

The encapsulation of Cur into Cur@PHBX-PR/FUdR_15S_ was examined by high-performance liquid chromatography (HPLC) equipped with the Agilent Extend C-18 column (4.6 × 250 mm, 5.0 μm) connected to a UV detector. A flow rate of 1.0 mL/min was used with a linear gradient of 4% glacial acetic acid solution (52%) and acetonitrile (48%). All samples were monitored at 425 nm. The drug loading of FUdR_15S_ labeled with FAM was detected by fluorescence spectrophotometry at 492 nm. The contents of FUdR_15S_ and Cur in nanodrugs were determined using an indirect method and calculated on the basis of their respective standard curves. The DL capacity of FUdR_15S_ and the entrapment efficiency (EE) of Cur were calculated in accordance with the following formulas:$$\;\mathrm{DL}\;(\%)=\frac{\mathrm{weight}\;\mathrm{of}\;\mathrm{Cur}/\mathrm{FUd}R_{15S}\;\mathrm{in}\;\mathrm{Cur}@\mathrm{PHBX}-\mathrm{PR}/\mathrm{FUd}R_{15S}}{\mathrm{weight}\;\mathrm{of}\;\mathrm{Cur}@\mathrm{PHBX}-\mathrm{PR}/\mathrm{FUd}R_{15S}\;}\;\times \;100\%\;\mathrm{EE}\;(\%)=\frac{\mathrm{weight}\;\mathrm{of}\;\mathrm{Cur}\;\mathrm{in}\;\mathrm{Cur}@\mathrm{PHBX}-\mathrm{PR}/\mathrm{FUd}R_{15S}}{\mathrm{weight}\;\mathrm{of}\;\mathrm{total}\;\mathrm{Cur}\;}\;\times \;100\%$$

The DL of Cur was determined using the direct method. Cur@PHBX-PR/FUdR_15S_ nanospheres (1 mg) were dissolved in 2 mL dichloromethane to release Cur from the nanospheres. The solution was diluted 10 times with methanol, and tested by HPLC. The DL of Cur was also calculated using the above formulas.

### 2.7. Stability Analysis of Cur@PHBX-PR/FUdR_15S_

Stability assays of Cur@PHBX-PR/FUdR_15S_ in BSA and different pH were conducted in various simulated in vivo conditions. Cur@PHBX-PR/FUdR_15S_ was incubated in 10% unheated FBS for 1, 2, 4, 8, and 12 h and in acidic (pH 4.5), neutral (pH 7.4), and basic (pH 8.0) buffer solutions at 37 °C for 2.0 h. Subsequently, these PHBX nanospheres were also incubated in acidic buffer (pH 4.5) with 20 U/mL DNase II for 0.25, 0.5, 1.0, 2.0, 4.0, and 8.0 h. Samples were loaded onto 2% agarose gel for electrophoresis under 100 V in TAE buffer. The gel was stained using the GelStain and visualized using the Tanon 2500 (Tanon Science & Technology Co., Ltd., Shanghai, China).

### 2.8. In Vitro Drug Release of Cur@PHBX-PR/FUdR_15S_

The *in vitro* drug release behaviors of Cur@PHBX-PR/FUdR_15S_ were determined using a dialysis bag (MWCO, 10 kDa). Cur@PHBX-PR/FUdR_15S_ solution (1 mL, 100 μM) was transferred into a dialysis bag and immersed into 10 mL acetate buffer (pH 4.5) with DNase II (20 U/mL) at 37 °C. Experiments were carried out in an incubator shaker (ZWYR–200D, LABWIT Scientific, Shanghai, China) with gentle shaking at a speed of 100 rpm. At predetermined time intervals, 0.5 mL release media was removed and replaced with 0.5 mL fresh release media. The concentrations of FUdR_15S_ and Cur in the solution were determined by fluorescence spectrophotometry and HPLC [32], respectively. The release rates of FUdR_15S_ and Cur were calculated using the following formulas:$$\mathrm{FUd}R_{15S}\;\mathrm{release}\;\mathrm{rate}\%=\frac{\mathrm{the}\;\mathrm{amount}\;\mathrm{of}\;\mathrm{cells}\;\mathrm{released}\;\mathrm{FUd}R_{15S}}{\mathrm{the}\;\mathrm{amount}\;\mathrm{of}\;\mathrm{FUd}R_{15S}\;\mathrm{in}\;\mathrm{the}\;\mathrm{carrier}}\times 100\%\;\mathrm{Cur}\;\mathrm{release}\;\mathrm{rate}\%\;=\frac{\mathrm{the}\;\mathrm{amount}\;\mathrm{of}\;\mathrm{cells}\;\mathrm{released}\;\mathrm{Cur}}{\mathrm{the}\;\mathrm{amount}\;\mathrm{of}\;\mathrm{Cur}\;\mathrm{in}\;\mathrm{the}\;\mathrm{carrier}}\times 100\%$$

The *in vitro* release curves of Cur@PHBX-PR/FUdR_15S_ were established.

### 2.9. Cell Cultures

Breast cancer cells (BT474) and cervical cancer cells (Hela) were grown in Gibco ^®^RPMI 1640 medium containing 10% FBS. Gastric cancer cells (MGC-803), liver cancer cells (HEPG2), and pancreatic cancer cells (Aspc-1) were grown in Gibco ^®^RPMI DMEM medium (containing 10% FBS). The above mentioned cells were cultured at 37 °C in a humidified atmosphere containing 5% CO_2_.

### 2.10. Evaluation of Cellular Uptake

The uptake of Cur@PHBX-PR/FUdR_15S_ by cells was qualitatively analyzed by laser confocal scanning electron microscopy. MGC 803 cells (1 × 105 cells/well) were separately seeded into a laser confocal dish (NEST, Wuxi, China) overnight at 37 °C. FAM-labeled Cur@PHBX-PR/FUdR_15S_ was used to treat cells for 0.5, 1, 2, 4, 8, and 16 h. The culture solution was carefully removed, and cells were washed thrice with ice-cold PBS and fixed with 4% formaldehyde for 20 min at room temperature. Then, cell nuclei were stained using 2.5 μg/mL DAPI for 30 min at 37 °C and rinsed with PBS thrice. Fluorescent images were obtained using the Zeiss laser scanning confocal microscope (Zeiss LSM 810, Oberkochen, Germany). The colocalization experiment was carried out using the lysosome staining reagent to illuminate the mechanism of cellular uptake of Cur@PHBX-PR/FUdR_15S_. The lysosome was stained with the LysoTracker Red for 0.5 h after MGC 803 cells were incubated with Cur@PHBX-PR/FUdR_15S_ for 1 h. Then, the image of the lysosome tracker signal was visualized using fluorescence microscopy with an excitation wavelength of 577 nm, and the Pearson’s correlation coefficient (r) was determined using image analysis software (Image Pro-Plus). All images were recorded, and target cells were counted using the 40× oil objective.

### 2.11. In Vitro Cytotoxicity

In vitro cytotoxicity assays of cancer cells were performed using the MTT method [33]. Briefly, exponentially growing MGC 803, BT474, Hela, HEPG2, and Aspc-1 cells were harvested and plated into 96-well plates at a concentration of 4 × 103 cells/well. After the cells were incubated at 37 °C for 24 h, the culture medium was replaced with 100 μL of fresh medium containing different concentrations of PHBX, FUdR, Cur, FUdR/Cur (2.3:1), and Cur@PHBX-PR/FUdR_15S_, and cells were incubated for an additional 48 h. Afterward, 10 μL MTT (5 mg/mL) was added into each well, and plates were incubated at 37 °C for 4 h. The supernatant was discarded, and 100 μL DMSO was added into each well. The absorbance was determined at 570 nm. Data were reported as mean of three independent experiments, and each run was done in quintuplicate. The dose–response graph was plotted by calculating the percent cell viability by using the formula below:$$\mathrm{Cell}\;\mathrm{viability}\;=\frac{{\mathrm{OD}}_{570\left(\mathrm{sample}\right)}-{\mathrm{OD}}_{570\left(\mathrm{blank}\right)}}{{\mathrm{OD}}_{570\left(\mathrm{control}\right)}-{\mathrm{OD}}_{570\left(\mathrm{blank}\right)}}\times \;100\%$$

In addition, the inhibitory concentrations causing 50% growth inhibition (IC50 value) of FUdR and Cur alone and in combination were determined using an online calculator (https://www.aatbio.com/tools/ic50–calculator accessed on 8 September 2020).

The combination index (CI) was calculated in accordance with the following formula:$$\mathrm{CI}\;=\frac{D_{\mathrm{FUdR}}}{{\left(D_{x}\right)}_{\mathrm{FUdR}}}+\frac{D_{\mathrm{Cur}}}{{\left(D_{x}\right)}_{\mathrm{Cur}}}$$

(*D_x_*)_FUdR_ and (*D_x_*)_Cur_ are the IC50 values of FUdR and Cur as a free drug, while *D*_FudR_ and *D*_Cur_ are the IC50 values of FUdR and Cur in Cur@PHBX-PR/FUdR_15S_, respectively, in combination at the given inhibition rate. CI values of <1, 1, and >1 represent synergism, additive, and antagonism, respectively.

### 2.12. Western Blot Analysis

MGC 803 cells were seeded on six-well plates (2 × 10^5^ cells/well) and cultured for 24 h. Then, FUdR, Cur, FUdR/Cur (2.3:1), and Cur@PHBX-PR/FUdR_15S_, were added into six-well plates (2 × 10^5^ cells/well), and the final concentrations of FUdR and Cur were 60 and 26 μM, respectively. After 24 h, culture media were discarded, and cells were washed with cold PBS buffer twice for harvest. Cell pellets were disrupted in the cell RIPA buffer (0.5% NP-40, 50 mM Tris-HCl, 120 mM NaCl, 1 mM EDTA, 0.1 mM Na_3_VO_4_, 1 mM NaF, 1 mM PMSF, and 1 μg/mL leupeptin; pH 7.5), and the protein concentrations were determined using the BCA kit. After SDS-polyamide gel electrophoresis, the eluent was then transferred onto PVDF membranes, which were blocked with 5% milk in TBST (5 mM Tris-HCl, 136 mM NaCl, and 0.1% Tween-20, pH 7.6) for 1 h. Membranes were cultured with primary antibodies against Bcl-2, Bax, and β-actin overnight at 4 °C, washed thrice with 1× TBST, incubated with horseradish peroxidase-conjugated secondary antibodies at room temperature for 1 h, and washed thrice with 1× TBST. Protein bands were visualized using the Tanon 5200 ECL system (Tanon Science & Technology Co., Ltd., Shanghai, China) and analyzed using the Image J software.

### 2.13. Caspase Activity Assays

MGC 803 cells (2 × 10^5^ cells/well) were inoculated into six-well plates and subsequently grouped the same way as described above for the Western blot experiment. Six-well plates were placed in an incubator for 10 (caspase 8 and 9 tests) or 24 (caspase 3 test) h. The cells of each well were collected and lysed for 15 min on ice in 50 μL lysis buffer provided by the kit. After centrifugation (18,000 rpm, 4 °C, 10 min), supernatants were collected. The protein concentration was measured using the BCA method. Substrates were added to the supernatant (adjusted final concentration to 2 mg/mL) and incubated at 37 °C for 4 h. The absorbance was measured at a wavelength of 405 nm on the Synergy HT microplate reader (Biotek Instruments, Inc., Green Mountains, VT, USA). Then, the relative caspase activity was calculated in accordance with the ratio of the absorbance values of apoptosis-induced and blank control cells.

### 2.14. Statistical Analysis

All samples were prepared and tested at least thrice. Data were presented as mean ± standard deviation. Significant differences among groups were determined using Newman–Keuls analysis. Differences were considered significant at * *p* < 0.05, highly significant at ** *p* < 0.01, and extremely significant for *** *p* < 0.001.

## 3. Results and Discussion

### 3.1. Preparation and Characterization of Cationic Peptide-Coated PHA Nanosphere

A new cationic peptide-coated PHA nanosphere was synthesized through multistep reactions (Figure 2). Firstly, a PHBHHx particle was prepared by emulsification/solvent evaporation. In this process, Cur was encapsulated in the hydrophobic hole of the particle, thus forming a hydrophobic core named Cur@PHBX. Secondly, fusion protein (PhaP–RALA, Appendix A), as a cationic and hydrophilic shell, was adsorbed on the surface of the hydrophobic core of Cur@PHBX under a strong hydrophobic effect. To more intuitively verify whether the hydrophilic core and the hydrophobic shell were successfully assembled, EGFP with green fluorescence and Rho with red fluorescence were used to replace RALA and Cur to complete the assembly process. The resulting assembly of Rho@PHBX-PE was detected using laser confocal microscopy (Figure 1a). The analytical results showed that Rho in the particles of Rho@PHBX emitted red fluorescence while the fusion protein of PhaP–EGFP produced green fluorescence under laser excitation. The successfully assembled nanosphere showed yellow fluorescence because of the overlap of the red light of Rho and green light of PhaP–EGFP. From this evidence, it was deduced that the expected assembly of Cur@PHBX-PR was achieved through the above process. Thirdly, we continued to perform the loading task of nucleic acid drugs on the surface of Cur@PHBX-PR. The addition of Cy5-labeled FUdR_15S_ to the solution of Cur@PHBX-PR promoted a strong electrostatic adsorption between negatively charged FUdR_15S_ and positively charged RALA, resulting in the formation of the assembled nanosphere (Cur@PHBX-PR/FUdR_15S_). Agarose gel retardation was utilized to characterize the formation of these assembled nanospheres. As illustrated in Appendix A, the free FUdR_15S_ strand migrated to the bottom of the gel while the assembled nanospheres in the sample wells showed no electrophoretic mobility.

In addition, the fluorescent image of assembled nanospheres of Cur@PHBX-PR/FUdR_15S_ were observed by a confocal microscope. A unique feature of orange fluorescence could be captured by overlapping the yellow fluorescence of Cur and red fluorescence of Cy5-labeled FUdR_15S_, which consistently demonstrated the formation of self-assembly of Cur@PHBX-PR/FUdR_15S_ (Figure 1b).

Furthermore, the morphologies of PHBHHx, Cur@PHBX, and Cur@PHBX-PR/FUdR_15S_ were observed. It was found that the particle size of pure PHBHHx was about 100 nm (Figure 2a) due to the addition of Tween-20 in the preparation of micro-emulsion. Application of this method could also control the size of Cur-loaded Cur@PHBX and nanospheres of Cur@PHBX-PR/FUdR_15S_ in the range of about 100 nm. TEM images revealed that the surface of Cur@PHBX-PR/FUdR_15S_ adsorbed with protein and FUdR_15S_ strand was rough and irregular compared with the surface of PHBX and Cur@PHBX (Figure 2b). Zeta potential analysis gave evidence that Cur@PHBX-PR/FUdR_15S_ was a cationic nanosphere with a ζ potential of +23.17 mA.

To determine the loading efficiency of the system, we detected and analyzed the loading capacity of FUdR and Cur by a fluorescence detector and HPLC. According to the standard curve of Cur, the amount of Cur loaded on 1 mg of Cur@PHBX-PR/FUdR_15S_ was 0.0779 mg (211.4 nmol), as depicted in Appendix A. The entrapment efficiency of Cur was 88.3% and the drug loading was 7.8%. FUdR was loaded with 489.5 nmol by 1 mg of Cur@PHBX-PR/FUdR_15S_, and the drug loading of FUdR was 12.1%. Compared with the drug loading of Cur and FUdR in the DNA micelle (Cur@affi-F/GQs) prepared by Zhang et al. [32], our drug loading of Cur was higher (5.5%), while drug loading of FUdR was lower than that of Zhang et al. (21.1%). Our results were better than those obtained by Iurciuc-Tincu, who prepared biocompatible polymeric micelles and co-loaded Cur and 5-FU with drug loading percentages of 6.4% and 5.8%, respectively [34].

### 3.2. Analysis of Stability of Cur@PHBX-PR/FUdR_15S_

Cur@PHBX-PR/FUdR_15S_ was incubated in solution with BSA at different pH values and DNase II treatment times to evaluate the stability. Processed samples were analyzed using 1% agarose gel electrophoresis. As shown in Figure 3a, Cur@PHBX-PR/FUdR_15S_ still maintained a stable structure in BSA even with prolonged incubation time. Similarly, under the neutral pH condition, Cur@PHBX-PR/FUdR_15S_ also displayed good stability. These results revealed that Cur@PHBX-PR/FUdR_15S_ had excellent stability and effectively prevented the destruction of their structures by serum albumin. However, partial FUdR_15S_ strands were detected at the bottom of the gel under acidic (pH 4.5) and alkaline (pH 9.0) conditions (Figure 3b). This might be attributed to the destruction of PhaP–RALA, causing the FUdR_15S_ to fall off from the nanospheres. In addition, after Cur@PHBX-PR/FUdR_15S_ was incubated with DNase II at pH 4.5 for 1 h, the structure of nanospheres was almost destroyed, and FUdR_15S_ was released. The dissociated FUdR_15S_ strand was completely degraded after incubation with DNase II for 24 h (Figure 3c). These phenomena also proved that the nanospheres were easily destroyed in the lysosome at pH 4.5, which was conducive to the release of Cur and FUdR.

### 3.3. In Vitro Drug Release

The *in vitro* release behavior of Cur@PHBX-PR/FUdR_15S_ was examined using a dialysis method. As displayed in Figure 3d, a sustained release was detected in the simulated physiological lysosome acidic conditions (pH 4.5). The cumulative release of Cur and FUdR_15S_ reached 35.5% and 86.7%, respectively, after 24 h of dialysis. The initial burst of FUdR_15S_ release could be explained by the statement that FUdR_15S_ was weakly loaded on the surface and was first released into the external environment when it came into contact with the acidic environment of the tumor [35]. However, under the protection of PHBHHx, the release of Cur from the core region to the external solution was relatively slow, mainly through the diffusion action of the drugs. With prolonged time, both drugs continued to be released stably. The release rate of FUdR_15S_ was 95.24% at 144 h, and the value of Cur was 89.56% at 384 h. The sustained-release time of FUdR and Cur from the PHBHHX nanosphere was longer than that from other polymer nanocarriers and the maximum release of the dual drug was also the highest [36,37,38]. The time-differential release of FUdR and Cur also increased the efficacy of the drugs in tumor cells. Additionally, no free FUdR was detected in the above process, which demonstrated that the FUdR_15S_ strand formed by FUdR monomer through phospholipid bonds was stable in different pH environments. Taken together, these results revealed that PHBHHX nanospheres have superior controlled release capabilities.

### 3.4. Cellular Uptake

The cellular uptake of Cur@PHBX-PR/FUdR_15S_ by gastric cancer MGC 803 cells (HER2 low-expression cell line) was monitored through confocal laser scanning microscopy. As depicted in Figure 4a, the red fluorescence of Cy5-labeled FUdR_15S_ and the yellow fluorescence of Cur were readily detected when MGC 803 cells were treated with Cur@PHBX-PR/FUdR_15S_. The weak fluorescence of FUdR and Cur in Cur@PHBX-PR/FUdR_15S_ could be captured at 0.5 h. As Cur@PHBX-PR/FUdR_15S_ was gradually absorbed into the cells, the fluorescence intensity increased. The maximum fluorescence intensity occurred at 2 h. With the increase in absorption time, the fluorescence intensity began to decrease. Only weak fluorescence could be detected at 16 h, suggesting that most of the drugs had been absorbed and metabolized by MGC 803 cells.

Cellular localization of Cur@PHBX-PR/FUdR_15S_ was further investigated. Figure 4b displays that green fluorescence from FAM-labeled FUdR_15S_ and yellow fluorescence of Cur were localized in lysosomes by using lysosome staining with LysoTracker Red. The co-localization levels of Cur@PHBX-PR/FUdR_15S_ with green/yellow fluorescence and lysosomes with red fluorescence were analyzed by calculating *r* through the Image Pro-Plus software (Appendix A). The *r* values of FUdR_15S_ and Cur of Cur@PHBX-PR/FUdR_15S_ in lysosomes were 0.93 and 0.81, respectively, which indicated that most of Cur@PHBX-PR/FUdR_15S_ was trapped in lysosomal compartments. This co-localization test further confirmed that Cur@PHBX-PR/FUdR_15S_ was internalized into MGC 803 cells via receptor-mediated endocytosis.

### 3.5. Studies on Cytotoxicity and Synergistic Effect

Firstly, it was proved that PHBX-PR was not toxic to a variety of cancer cells (i.e., MGC 803, BT474, HepG2, AsPC-1, and HeLa cells) by using MTT methods (Appendix A), uncovering the biocompatibility and safety of PHBX-PR as a drug delivery carrier. Subsequently, the antitumor effects of free FUdR, Cur, FUdR/Cur (2.3:1, a physical mixture of FUdR and Cur at a molar ratio of 2.3:1), and Cur@PHBX-PR/FUdR_15S_ on different cancer cells were detected. A concentration-dependent increase in cytotoxicity was observed in MGC 803 cells treated with the drugs above (Figure 5a). Compared with FUdR and Cur alone, FUdR/Cur (2.3:1) and Cur@PHBX-PR/FUdR_15S_ exhibited higher cell cytotoxicity. The IC50 values of FUdR and Cur were 16.0 and 2.5 μM, respectively, which were the lowest in all treatment groups (Figure 6). The synergetic effect of FUdR and Cur was quantified through the analysis of CI, as previously described [33]. According to Appendix A, the CI value of Cur@PHBX-PR/FUdR_15S_ was lower than that of physical combination (0.188 vs. 0.888), which showed a remarkable synergistic anticancer effect of FUdR and Cur in Cur@PHBX-PR/FUdR_15S_. This synergistic effect was also consistent with that of combined administration of the two drugs in other carriers [39]. Such a synergistic effect depends on Cur to enhance the anticancer efficacy by increasing the cytotoxicity of FUdR [40].

In addition, it was found that BT474, Aspc-1, HEPG2, and Hela cells treated with these drugs had similar trends (Figure 5b–e). The assembled drug of Cur@PHBX-PR/FUdR_15S_ had better inhibitory and synergistic effects than the physical combination of FUdR/Cur (2.3:1) and free FUdR and Cur. The IC50 values of FUdR and Cur in Cur@PHBX-PR/FUdR_15S_ were 25.54 ± 4.308 and 11.11 ± 4.334 μM, respectively, in BT474; 19.77 ± 4.219 and 8.609 ± 4.206 μM, respectively, in Aspc-1; 40.41 ± 2.902 and 17.58 ± 2.904 μM, respectively, in HEPG2; and 21.83 ± 4.486 and 9.506 ± 4.506 μM, respectively, in Hela (Figure 6, Appendix A). These results also supported the view that the IC50 value of the dual-drug-loaded composite is lesser than that of single drug or of dual drugs in physical mixture [41]. The co-delivery of FUdR and Cur by cationic peptide-coated PHBHHx nanospheres was suitable for different cancer cells and improved the coordinated therapeutic effect of the two drugs, even though the sensitivity of different cells to the drug concentration varied markedly.

### 3.6. Exploration of Anticancer Mechanism

Cell apoptosis assays were conducted using Annexin V-FITC/propidium iodide double staining to verify the mechanism of enhanced antitumor activity by Cur@PHBX-PR/FUdR_15S_. As displayed in Figure 7a, the apoptosis rate of MGC 803 induced by free FUdR was 28.18%, and the value induced by Cur alone was 19.65%. However, cells treated with combination drugs, e.g., FUdR/Cur (2.3:1) and Cur@PHBX-PR/FUdR_15S_, raised the apoptosis rate significantly. The trend of apoptosis was the most evident when the cells were treated with Cur@PHBX-PR/FUdR_15S_, and the apoptosis rate was as high as 93.52%. Cur and FUdR in cationic peptide-coated PHBHHx nanospheres had a remarkable synergistic effect on cancer cells via the mechanism of enhanced apoptosis. These results were consistent with the results of other studies such as those of Zheng et al., who indicated that Cur enhanced the apoptosis induced by 5-FU [42].

Furthermore, to reveal the mechanism of anticancer effects of nano-drugs, the expression and activity of apoptosis-related proteins in MGC 803 cells treated with free FUdR, Cur, FUdR/Cur (2.3:1), and Cur@PHBX-PR/FUdR_15S_ were detected by Western blot and caspase activity assay kits. As shown in Figure 7b, Cur@PHBX-PR/FUdR_15S_ led to evident downregulation of the expression of Bcl-2 and upregulation of the expression of Bax compared with the control group (PBS) or the groups treated with free FUdR, Cur, or FUdR/Cur (2.3:1). The Bax/Bcl-2 ratio in Cur@PHBX-PR/FUdR_15S_-treated cells was 5.74, which was about 3- and 4-fold higher than those in FUdR/Cur- and FUdR-treated cells (1.75 and 1.39, respectively; Figure 7c). It is generally agreed upon that FUdR lead cancer cells to enter the apoptotic pathway via the mitochondrial pathway regulated by the Bcl-2 protein family [43]. Our results also confirmed that the expression levels of Bcl-2 and Bax had a significant effect on cell death.

Moreover, the caspase protease family contains key enzymes that induce apoptosis, and the activities of caspases 3, 8, and 9 are also regulated by the Bcl-2 protein family [44]. The results of caspase activity (Figure 7d) indicated that caspases 3, 8, and 9 were of high activity in cells treated with Cur@PHBX-RP/FUdR_15S_, whereas low activity was found in cells treated with free FUdR, Cur, and FUdR/Cur (2.3:1). To sum up, we propose the anticancer mechanism of combined FUdR and Cur; that is, Cur improves the activity of apoptosis-related proteins and sensitizes gastric cancer cells to FUdR. As a result, the downregulation of anti-apoptosis protein Bcl-2, upregulation of apoptosis protein Bax, and the enhancement of activities of caspases 3, 8, and 9 lead to the increase in apoptosis rate of cancer cells. During this process, cationic peptide-coated PHBHHx nanospheres, as drug carriers, play a significant role in promoting the synergistic effect of FUdR and Cur.

## 4. Conclusions

In the present study, new amphiphilic cationic peptide-coated PHA nanospheres were prepared using PHBHHx as the hydrophobic core and cationic peptide RALA as the hydrophilic shell. Based on this carrier, hydrophobic drugs of Cur were encapsulated into the core, and nucleic acid drugs of FUdR_15S_ strand were absorbed by the shell. These novel amphiphilic cationic vesicles were proven to be capable of loading large quantities of drugs and releasing drugs rapidly at acidic conditions with DNase II. Furthermore, antitumor experiments confirmed that Cur@PHBX-PR/FUdR_15S_ remarkably improved the anticancer efficacy and played a vital role in promoting the synergistic effect of FUdR and Cur. Moreover, it was inferred from the exploration into anticancer mechanisms that Cur enhanced the activity of apoptosis-related proteins and sensitized gastric cancer cells to FUdR. Such multifunctional amphiphilic cationic polymer vesicles can provide a platform for co-delivering hydrophobic drugs and nucleic acid drugs, genes or CRISPR, and even proteins to treat cancer.

## Data Availability

Not applicable.

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
