# Peer review of "Amphiphilic Cationic Peptide-Coated PHA Nanosphere as an Efficient Vector for Multiple-Drug Delivery"

_nanomaterials, 2022, doi:10.3390/nano12173024_

Round 1

Reviewer 1 Report

The manuscript proposed by Fanghua Zhang et al., aims to develop Amphiphilic Cationic Peptide-coated PHA Nanosphere in order to deliver different therapeutic drugs.

The manuscript is easy to read, however there are some major comments.

-The manuscript lacks a real “Discussion”: among the 37 references given, 31 are in the “Introduction” part , 3 are in the “Material and Methods” part, and only 3 references are in “Results and Discussion”. In the "Results and Discussion" section, the authors should discuss their results in relation to the literature: what is new, why it is better, the % of drug encapsulation in relation to other types of NPs, etc.

- it seems that the reference 34 does not appear in the text..

Minor comments:

 Scheme 1 : error in writing "nuclous" correct is "nucleus

Scheme 1 : is too small and illegible, write larger please

Figure 1: The figure has a title but no legend (which describes the figure)

Figure2 : the text is not readable

Figure 3 : the text is not readable. Please add a legend for the figure (not only the title)

Figure4 : Enlarge the figure since is not readable.  The Figure 4a) : the bright field does not show any cells.

Figure 5-6-7: The text is unreadable. There is a title but no legend. Also, no statistical test appears on any figure.

Author Response

Dear Reviewer:

Thank you for your advice.

Our manuscript has been revised as requested and submitted for your further consideration. All the questions and issues in comments are answered and highlighted in yellow color in the manuscript as well as described in the below.

I wish this revised version would meet the requirement for publication in the journal. If there is any question or request, please feel free to contact me. We will modify the manuscript until it is acceptable. Your help is highly appreciated.

Quention 1:

The manuscript is easy to read, however there are some major comments.

-The manuscript lacks a real “Discussion”: among the 37 references given, 31 are in the “Introduction” part , 3 are in the “Material and Methods” part, and only 3 references are in “Results and Discussion”. In the "Results and Discussion" section, the authors should discuss their results in relation to the literature: what is new, why it is better, the % of drug encapsulation in relation to other types of NPs, etc.

Answer:

We have added discussions comparing with other literatures to the Results and Discussion Section. The detailed descriptions are highlighted in yellow.

Quention 2:

It seems that the reference 34 does not appear in the text..

Answer:

We have reordered the references and added some new ones in the Results and Discussion section. The revised references are highlighted in red.

Quention 3:

Scheme 1 : error in writing "nuclous" correct is "nucleus

Answer:

We have modified the error writing in Scheme 1.

Quention 4:

Scheme 1 : is too small and illegible, write larger please.

Answer:

We have wrote larger of the text on the Scheme 1.

Quention 5:

Figure 1: The figure has a title but no legend (which describes the figure).

Answer:

The legend have been added for the Figure 1.

Quention 6:

Figure2: the text is not readable

Answer:

We have modified the Figure 2.

Quention 7:

Figure 3 : the text is not readable. Please add a legend for the figure (not only the title)

Answer:

We have changed the picture, and added a legend for the Figure 3.

Quention 8:

Figure4 : Enlarge the figure since is not readable.  The Figure 4a) : the bright field does not show any cells.

Answer:

The Figure 4 has been modified so that it can be read easily.

Quention 9:

Figure 5-6-7: The text is unreadable. There is a title but no legend. Also, no statistical test appears on any figure.

Answer:Three of the figures have been modified and the legends have been added. The statistical test have been added too.

Reviewer 2 Report

The authors presented a new cationic peptide-coated PHA nanosphere to coload nucleic acid drugs and hydrophobic drugs. A number of experiments have been carried out to confirm the desired properties of the obtained product and its safety. 

The study was well organized and carried out. I have no major remarks. The only issue to which I would like to point the authors' attention to is the proper font size on the figure - e.g. on figures 3, 5, 6, and 7 it is too small.

Author Response

Dear Reviewer:

Thank you for your advice.

Our manuscript has been revised as requested and submitted for your further consideration. All the questions and issues in comments are answered and highlighted in yellow color in the manuscript as well as described in the below.

Quention:

The only issue to which I would like to point the authors' attention to is the proper font size on the figure - e.g. on figures 3, 5, 6, and 7 it is too small.

Answer:

We have enlarged the front size of the text of all the figures above.